# The Intestinal Microbiome in Dogs with Chronic Enteropathies and Cobalamin Deficiency or Normocobalaminemia—A Comparative Study

**DOI:** 10.3390/ani13081378

**Published:** 2023-04-17

**Authors:** Linda Toresson, Jan S. Suchodolski, Thomas Spillmann, Bruna C. Lopes, Johnathan Shih, Jörg M. Steiner, Rachel Pilla

**Affiliations:** 1Evidensia Specialist Animal Hospital Helsingborg, 254 66 Helsingborg, Sweden; 2Department of Equine and Small Animal Medicine, Faculty of Veterinary Medicine, University of Helsinki, 00014 Helsinki, Finland; thomas.spillmann@helsinki.fi; 3Gastrointestinal Laboratory, Department of Small Animal Clinical Sciences, Texas A&M University, 4474 TAMU, College Station, TX 77843-4474, USA; jsuchodolski@cvm.tamu.edu (J.S.S.); bclopes@cvm.tamu.edu (B.C.L.); jbs@tamu.edu (J.S.); jsteiner@cvm.tamu.edu (J.M.S.); rpilla@cvm.tamu.edu (R.P.)

**Keywords:** chronic diarrhea, vitamin B12, microbiota, canine

## Abstract

**Simple Summary:**

Cobalamin is a vitamin that all cells of humans and animals, as well as bacteria, need to survive. If the intestine of dogs is diseased, serum cobalamin levels can decrease in some of these dogs. There are no studies in dogs using modern techniques to compare the gut bacteria in dogs with chronic gut inflammation with or without low serum cobalamin levels. Therefore, we compared the gut bacteria in 47 dogs with chronic gut inflammation. Twenty-nine of them had a low serum cobalamin level, while 18 did not. We found that those dogs with a decreased serum cobalamin level had severe alterations in the composition of their intestinal bacteria, while those with a normal serum cobalamin level did not. Oral or injectable supplements did not correct the changes in intestinal bacteria, suggesting that low serum cobalamin levels are an indicator of changes in intestinal bacteria rather than their cause.

**Abstract:**

Cobalamin deficiency is a common sequela of chronic enteropathies (CE) in dogs. Studies comparing the intestinal microbiome of CE dogs with cobalamin deficiency to those that are normocobalaminemic are lacking. Therefore, our aim was to describe the fecal microbiome in a prospective, comparative study evaluating 29 dogs with CE and cobalamin deficiency, 18 dogs with CE and normocobalaminemia, and 10 healthy control dogs. Dogs with cobalamin deficiency were also analyzed after oral or parenteral cobalamin supplementation. Overall microbiome composition (beta diversity) at baseline was significantly different in CE dogs with cobalamin deficiency when compared to those with normocobalaminemia (*p* = 0.001, R = 0.257) and to healthy controls (*p* = 0.001, R = 0.363). Abundances of Firmicutes and Actinobacteria were significantly increased (q = 0.010 and 0.049), while those of Bacteroidetes and Fusobacteria were significantly decreased (q = 0.002 and 0.014) in CE dogs with cobalamin deficiency when compared to healthy controls. Overall microbiome composition in follow-up samples remained significantly different after 3 months in both dogs receiving parenteral (R = 0.420, *p* = 0.013) or oral cobalamin supplementation (R = 0.251, *p* = 0.007). Because cobalamin supplementation, in combination with appropriate therapy, failed to restore the microbiome composition in the dogs in our study, cobalamin is unlikely to be the cause of those microbiome changes but rather an indicator of differences in underlying pathophysiology that do not influence clinical severity but result in a significant aggravation of dysbiosis.

## 1. Introduction

Cobalamin deficiency is a prevalent sequela in dogs with chronic enteropathies (CE) [1,2,3,4]. Once cobalamin deficiency has developed, the prognosis for a good response to treatment of the underlying condition is worse than in CE dogs without cobalamin deficiency [1,4]. The suggested mechanisms behind cobalamin deficiency in dogs with CE is a decreased expression of the cubam receptors in the ileum, causing impaired cobalamin absorption, hypocobalaminemia, and intestinal dysbiosis, which can cause bacterial competition for nutrients and less cobalamin being available for absorption in the ileum [5].

Mammalian cells require cobalamin as a cofactor for two enzymatic processes, but bacteria need cobalamin for over a dozen enzymatic processes [6,7]. A study has shown that 83% of bacteria present in the human gut microbiome encode cobalamin-dependent enzymes; however, most of those species were unable to synthesize cobalamin themselves [6]. The cobalamin biosynthetic pathway is present in all Fusobacteria, but it is rare in the phyla Actinobacteria and Proteobacteria, and only about half of Bacteroidetes encode it [8]. Therefore, those bacterial species need mechanisms to acquire cobalamin, and a two-step process with passive absorption and active transport has been described [9]. Interestingly, however, the avidity of bacteria for cobalamin varies, and in the recovery of cobalamin by the intrinsic factor (IF) secreted by the host is variable, depending on the bacterial species [10]. In a study by Degnan et al. [6], loss of cobalamin transporters was a disadvantage, resulting in decreased amounts and competitive ability of affected bacterial strains in the small intestine. Bacterial genes are also regulated by cobalamin through mechanisms such as riboswitches and others [11]. Therefore, the composition of the intestinal microbiome can reasonably affect the host’s ability to absorb cobalamin from foods, and oral cobalamin supplementation can confer a selective advantage for some bacterial species.

In a recent systematic review of the link between cobalamin and the gastrointestinal microbiome in people, results from in vitro studies suggested that cobalamin supplementation increases alpha diversity and affects beta diversity [12]. Results from human and animal studies were, however, discordant. Cobalamin supplementation increased alpha diversity in adults but not in infants or children [13,14,15,16]. Similar results were found regarding beta diversity. Furthermore, different forms of cobalamin (adenosylcobalamin, cyanocobalamin, and methylcobalamin) used for supplementation as well as co-intervention, appeared to impact the microbiome in different ways [17,18,19]. One study in mice showed that cobalamin supplementation was associated with significantly altered beta diversity but not alpha diversity [20], whereas beta diversity was unaffected in three other studies in mice [17,21,22]. At the genus level, an abundance of Bacteroides has been reported to decrease with oral supplementation with cyanocobalamin [20], a concerning finding since Bacteroides is commonly already decreased in dogs with CE and expected to improve with treatment [23,24,25].

The effects of oral versus parenteral cobalamin supplementation on the microbiome were not compared in any of the studies reviewed above [12]. Theoretically, oral cobalamin supplementation could create a gut environment with increased amounts of unbound cobalamin, while parenteral cobalamin supplementation would bypass the microbiome-host competition in the gut. Therefore, parenteral supplementation could potentially avoid the selective enrichment of cobalamin-dependent bacteria that are less capable of absorbing or transporting cobalamin in the gut.

Small intestinal bacterial overgrowth (SIBO), diagnosed with a culture of duodenal juice, was associated with cobalamin deficiency and increased serum folate concentrations in one study in dogs [26]. The combination of subnormal serum cobalamin concentrations and supranormal folate concentrations was only moderately sensitive and specific for canine SIBO in a later study using the same methodology [27]. In other studies, no or poor correlation between hypocobalaminemia and SIBO, proven with a quantitative culture of duodenal juice, has been found in dogs with SIBO and antibiotic-responsive diarrhea [3,28]. Using the search engines PubMed, Google Scholar, and Reef Seek, no direct evidence of causative organisms clearly linking dysbiosis to canine cobalamin malabsorption has been published in dogs to date. Furthermore, no studies in dogs using molecular tools, such as 16S rRNA or PCR, that specifically address the intestinal microbiome in dogs with hypocobalaminemia and CE were found.

Intestinal dysbiosis is a common sequel to CE in dogs [29,30]. However, the degree of dysbiosis that differs between CE dogs with and without hypocobalaminemia has not been reported based on searches using the previously mentioned search engines.

Therefore, the primary objective of this study was to characterize and compare the intestinal microbiome in CE dogs with cobalamin deficiency to those that are normocobalaminemic. The secondary objectives were to describe the effects of cobalamin supplementation on the microbiota and compare the effects of oral versus parenteral cobalamin supplementation on the intestinal microbiome in CE dogs with cobalamin deficiency.

## 2. Materials and Methods

### 2.1. Animal Inclusion

Three groups of dogs were included and classified based on clinical signs and laboratory exams: dogs with CE and low serum cobalamin concentrations, dogs with CE that were normocobalaminemic, and healthy controls. Dogs within both CE groups (regardless of cobalamin status) had to have clinical signs of chronic gastrointestinal disorders, such as diarrhea, weight loss, vomiting, and hyporexia, for a minimum of three weeks. Exclusion criteria for all groups of dogs were antibiotic treatment within 3 months prior to collection of fecal samples, being fed a raw food diet, concurrent exocrine pancreatic insufficiency, or being under treatment with a proton pump inhibitor.

The group of dogs with CE and low serum cobalamin concentrations included dogs with low normal serum cobalamin concentration (180–210 pmol/L; reference interval: 180–708 pmol/L) and dogs with hypocobalaminemia (serum cobalamin concentration <180 pmol/L). All of these dogs were also evaluated for evidence of intracellular cobalamin deficiency (i.e., supranormal serum methylmalonic acid (MMA) concentrations at baseline or MMA concentrations within reference interval at baseline that decreased significantly after one to three months of cobalamin supplementation) [31]. A reduction of MMA after cobalamin supplementation has been used in people to confirm pretreatment cobalamin deficiency in people with equivocal serum cobalamin and/or MMA concentrations [32,33,34]. Dogs at the lowest end of the serum cobalamin reference interval (180–210 pmol/L) were only included in this group if evidence of intracellular cobalamin deficiency was present. For simplicity, all dogs with low or subnormal serum cobalamin concentration and either supranormal serum MMA concentrations at baseline or a significant decrease in serum MMA concentration after cobalamin supplementation will be referred to as dogs with cobalamin deficiency.

Dogs with CE that were included in the normocobalaminemic group had serum cobalamin concentrations at or above the 33rd percentile of the lowest end of the reference interval (≥350 pmol/L). Dogs in this group were excluded if they were under treatment with cobalamin supplementation or had been supplemented with cobalamin within the last 12 months.

The healthy dog group included dogs that were clinically healthy and were volunteered by their owners as potential donors for a fecal microbiota transplant. All healthy controls had normal serum cobalamin concentrations.

### 2.2. Study Design, Baseline Data, and Diagnostic Investigations

This was an open, prospective comparative study. Fecal samples from all dogs were collected at the Evidensia Specialist Animal Hospital, Helsingborg, Sweden (ESAHHS) between March 2014 and October 2019 after informed owner consent. The study was approved by the Animal Ethics Committee in Uppsala (approval number C109/13; date of approval 27 September 2013). All fecal samples were freely passed. All CE dogs with cobalamin deficiency were participating in a block-randomized study of oral versus parenteral cobalamin supplementation [31,35]. From this group of dogs, fecal samples were collected at baseline and 90 +/− 15 days after cobalamin supplementation was initiated. From all other dogs, fecal samples were collected at one single time point. Samples from CE dogs with normocobalaminemia were collected from feces brought by the dog owners to the ESAHHS for other analyses, typically screening for fecal parasites. All of the samples from healthy dogs were collected from staff-owned dogs that were screened to potentially become fecal microbiota transplantation donors.

Fecal samples were refrigerated within 2 h of collection, frozen at −20 °C within 1–2 days, and stored at the laboratory at ESAHHS. Every 6–12 months, the frozen samples were sent on dry ice from ESAHHS to the Gastrointestinal Laboratory at Texas A&M University, College Station, Texas, using express delivery, and the conditions of the samples at arrival were reported.

With the exception of antibiotics and proton pump inhibitors, concurrent medication was allowed based on the treating clinician’s assessment. Dietary and medical history were collected at the time of inclusion. Intestinal parasites were excluded, and ultrasonography was performed to exclude extra-intestinal causes of clinical signs. Clinical data and work-up from dogs with cobalamin deficiency have been previously published [31]. Intestinal biopsies to confirm chronic inflammation were available from 23/29 dogs with CE and cobalamin deficiency and 11/18 dogs with CE and normocobalaminemia. Canine Inflammatory Bowel Disease Index (CIBDAI) was calculated at the time of consultation [36].

### 2.3. Serum Cobalamin and Methylmalonic Acid Concentrations

All serum samples were refrigerated within 2 h of collection and frozen at −20 °C for a minimum of 24 h prior to transport. Samples for cobalamin analysis were sent to the Laboratory Department at Evidensia Specialist Animal Hospital, Strömsholm, Sweden, with cold packs using priority delivery. Previous reports have shown stable serum cobalamin concentrations under similar conditions [37]. The samples were analyzed using an automated chemiluminescence immunoassay (Immulite 2000, Siemens Healthcare Diagnostics), and the detection limit was 110 pmol/L. Serum samples for MMA analysis were sent on dry ice every 6 to 12 months to the Gastrointestinal Laboratory at Texas A&M University, College Station, Texas, using express delivery. The samples were analyzed using a stable isotope dilution gas chromatography-mass spectrometry method, as previously described [35,38].

### 2.4. Microbiome Analysis

DNA was extracted from an aliquot of 100 mg of feces with a commercially available kit following the manufacturer’s instructions (PowerSoil^®^ DNA Isolation Kit, MOBIO Laboratories, Inc., Carlsbad, CA, USA). Sequencing of the V4 region of the 16S rRNA gene was performed at MrDna Laboratory (Molecular Research LP, Mr DNA, Shallowater, TX, USA) using primers 515F (5′-GTGYCAGCMGCCGCGGTAA) [39] to 806RB (5′-GGACTACNVGGGTWTCTAAT) [40]. Briefly, amplification was performed under the following conditions: 95 °C for 5 min, followed by 30 cycles of 95 °C for 30 s, 53 °C for 40 s, and 72 °C for 1 min, and a final elongation step at 72 °C for 10 min. After amplification, PCR products were checked by electrophoresis on a 2% agarose gel. Samples were then multiplexed using unique dual indices and pooled together in equal proportions based on their molecular weight and DNA concentrations. Pooled samples were purified using calibrated Ampure XP beads, and an Illumina DNA library was prepared. Sequencing was performed on a MiSeq following the manufacturer’s guidelines. The raw sequences were uploaded to NCBI Sequence Read Archive under accession number PRJNA863651.

Sequences obtained were processed using Quantitative Insights Into Microbial Ecology 2 (QIIME 2, v 2021.2) [41]. The sequence data were demultiplexed, and an amplicon sequence variant (ASV) table was created using DADA2 [42]. Sequences assigned as chloroplast, mitochondria, and low abundance ASVs (not present in at least 50% of samples from at least one group or time point) were removed prior to downstream analysis. To normalize sequencing depth across all samples, rarefaction was performed to a depth of 4990 sequences per sample, which was chosen based on the lowest read depth.

Alpha diversity was calculated using Chao 1, Shannon diversity, and observed species metrics. Beta diversity was evaluated by a weighted UniFrac distance matrix and visualized using PCoA (Principal Coordinate Analysis) plots.

### 2.5. Statistical Analysis

Prism 6.0 (GraphPad Software) was used for comparative data analyses of serum cobalamin, MMA, and CIBDAI. Normality testing was performed with the D’Agostino and Pearson omnibus normality test. Since the groups were not normally distributed, the Mann–Whitney test was used for all comparisons except serum MMA concentrations before and after cobalamin supplementation, in which the Wilcoxon matched-pairs signed rank test was used. Statistical significance was set as a *p*-value < 0.05.

Multivariate analysis was performed on the weighted UniFrac distance matrixes using the ANOSIM (Analysis of Similarity) test within PRIMER 7 software (PRIMER-E Ltd., Luton, UK) to analyze differences in microbial communities. Univariate analysis of bacterial taxa and alpha diversity was performed on Prism v.9.0 (GraphPad Software). Kruskall–Wallis test was used to compare groups at baseline and to compare follow-ups with healthy controls and adjusted at each taxonomic level for multiple comparisons using Benjamini and Hochberg’s False Discovery Rate [43]. A q-value < 0.05 was considered statistically significant. Group differences in bacterial taxa were determined with post hoc Dunn’s multiple comparison test.

## 3. Results

### 3.1. Baseline Data and Clinical Diagnosis

Fifty-seven dogs were included, of which 29 dogs had CE and cobalamin deficiency, 18 dogs had CE and were normocobalaminemic, and 10 dogs were healthy control dogs. Baseline data is available in Table 1. Thirty-one different breeds were included, of which the most common breeds were mixed breed dogs (10/57 (18%)), Labrador Retrievers (7/57 (12%)), German Shepherds (3/57 (5%)) and Golden Retrievers (3/57 (5%)). The remaining 27 breeds were represented by 1–2 dogs. An Australian shepherd, a breed with a predisposition for congenital cobalamin deficiency, was included in the group of normocobalaminemic CE dogs [44]. No other breed known for congenital cobalamin deficiency was included. In the group with cobalamin deficiency, 24/29 (83%) dogs had immune-suppressant responsive enteropathy (IRE), of which four dogs also had protein-losing enteropathy (PLE) with a serum albumin concentration below 20 g/L at baseline, 2/29 (7%) dogs had food-responsive enteropathy (FRE), 2/29 (7%) dogs had non-responsive enteropathy, of which one had PLE, and 1/29 dogs (3%) had antibiotic-responsive enteropathy (ARE). In the group of CE dogs with normocobalaminemia, 12/18 (67%) had IRE, 3/18 (17%) had FRE, 2/18 (11%) had NRE, and 1/18 (6%) was initially poorly responsive to immunosuppressive treatment but responded to fecal microbiota transplantation (FMT) and was generally stable on immunosuppressants after four FMTs. No dogs with PLE were present in this group. The clinical diagnosis was established over the first 6 months after inclusion based on treatment response.

### 3.2. Serum Cobalamin and Methylmalonic Acid Concentrations

Serum cobalamin concentrations (reference interval 180–708 pmol/L) were <111–210 pmol/L (median 183) in the cobalamin deficient group, 350–>738 pmol/L (median 492) in the CE group with normocobalaminemia, and 350–658 pmol/L (median 475) in the healthy control group. The cobalamin concentration was significantly lower in the cobalamin-deficient group compared to the group of CE dogs with normocobalaminemia (*p* < 0.001) and the healthy control group (*p* < 0.001). There was no significant difference in serum cobalamin concentrations between the healthy control group and the CE group with normocobalaminemia (*p* = 0.56).

Serum methylmalonic acid (MMA) concentrations were available from dogs with cobalamin deficiency and the healthy control group. Serum MMA concentrations (reference interval 415–1193 nmol/L) were 566–2468 nmol/L (median 934) in the cobalamin deficient group at baseline, which was significantly higher than the serum MMA concentrations of 470–972 nmol/L (median 746) in the healthy control group (*p* = 0.012). After cobalamin supplementation, the serum MMA concentration decreased significantly to 450–1221 nmol/L (median 626, *p* < 0.001). At this time point, there was no longer any significant difference in serum MMA concentrations between the healthy control group and the previously cobalamin-deficient group (*p* = 0.17). All the dogs with cobalamin deficiency either had supranormal MMA at inclusion or a significant reduction of serum MMA concentrations after 1–3 months of cobalamin supplementation.

Serum samples for MMA analysis from the normocobalaminemic dogs with CE were lost in a transatlantic shipment. One of the cardboard and corresponding styrofoam boxes with dry ice fell apart during the shipment, and the content was lost.

### 3.3. Cobalamin Supplementation, Concurrent Medication, and Diet

Of the 29 dogs with cobalamin deficiency, 18 dogs were treated with oral cyanocobalamin supplementation and 11 with parenteral hydroxocobalamin supplementation according to a previously described protocol [31]. At inclusion, 8/29 dogs were under treatment with immunosuppressant drugs in this group of dogs (Table 1). During the cobalamin supplementation study, 19 additional dogs were started on immunosuppressive treatment. In the group of CE dogs with normocobalaminemia, 10/18 (56%) dogs were under treatment with immunosuppressive drugs at inclusion (Table 1). Miscellaneous treatments have been listed in Table 1. In the cobalamin deficiency group, 22/29 (76%) dogs were fed kibbles from major pet food companies (Table 1). Two (7%) dogs were fed kibbles mixed with a homecooked diet, and 5/29 (17%) dogs were fed a homecooked, meat-based diet. None of these dogs were fed a raw food diet. One of the dogs was fed a balanced home-cooked diet according to the Association of American Feed Control Official (AAFCO) guidelines. In the group of dogs with CE and normocobalaminemia, as well as the healthy control group, all dogs were fed kibbles from major pet food companies.

### 3.4. Canine Inflammatory Bowel Disease Activity Index

There was no significant difference in CIBDAI between CE dogs with cobalamin deficiency and those normocobalaminemic at inclusion (*p* = 0.99). The CIBDAI range was 1–14 (median 7) in the cobalamin-deficient group at baseline and 5–11 (median 7) in the normocobalaminemic group. The CIBDAI range in the healthy control group was 1–2 (median 1), which is considered clinically insignificant [36]. This was significantly lower than CIBDAI in dogs with CE in the cobalamin deficient or normocobalaminemic group (*p* < 0.0001 for both groups, Figure 1).

Dogs with cobalamin deficiency were randomized to receive supplementation by either oral (PO) or parenteral (PE) route. No difference was observed between groups at baseline (*p* = 0.273) or at 3 months (*p* = 0.580). CIBDAI scores decreased significantly in both groups at 3 months compared with baseline (PO *p* < 0.0001, PE *p* = 0.0006).

### 3.5. Microbiome Analysis

Fifty-seven fecal baseline samples were available from all groups of dogs. Follow-up fecal samples after cobalamin supplementation were available from 20 dogs, of which 15 were in the oral supplementation group, and 5 were in the parenteral supplementation group. Reasons for lack of a follow-up sample were drop-out from the study for unknown reasons (2/9), euthanasia due to poor response to medical treatment (2/9), the dog owner forgetting to bring a fecal sample (2/9), or the dogs starting treatment with antibiotics (2/9) or proton-pump inhibitor (1/9) prior to follow-up.

At baseline, bacterial richness (Chao1) and evenness (Shannon Index) was reduced in dogs with CE and cobalamin deficiency (*p* = 0.013 and 0.043, respectively) but not in CE dogs with normocobalaminemia compared to healthy controls (Appendix A). No difference in Observed ASVs (another parameter of richness) was observed (Appendix A).

Overall microbiome composition (beta diversity, measured with Weighted UniFrac distances) at baseline was significantly different in CE dogs with cobalamin deficiency when compared to those with normocobalaminemia (*p* = 0.001, R = 0.257) and to healthy controls (*p* = 0.001, R = 0.363), as shown in Figure 2. No difference was found between CE dogs with normocobalaminemia and healthy controls (*p* = 0.976, R = −0.118).

When individual bacterial taxa were analyzed, major changes were observed at the phyla level (Figure 3), with 4/5 phyla significantly different from healthy controls in the cobalamin-deficient group (Appendix A). Firmicutes and Actinobacteria significantly increased (HC median 22.95 and 0.43%, cobalamin deficient median 69.03 and 3.12%, respectively), while Bacteroidetes and Fusobacteria significantly decreased (HC median 27.92 and 32.61%, cobalamin deficient median 0.19 and 0.54%, respectively) in cobalamin-deficient dogs. No significant changes were observed at the phyla level in CE dogs with normocobalaminemia compared to healthy controls.

At the genus level, cobalamin deficiency was associated with significant decreases in the abundance of Bacteroides (HC median 22.61%, CE cobalamin deficient 0.08%, q = 0.004), Sutterella (HC median 1.43%, CE cobalamin deficient 0.02%, q = 0.041), Helicobacter (HC median 0.64%, CE cobalamin deficient 0.00%, q = 0.004) and Turicibacter (HC median 0.28%, CE cobalamin deficient 0.00%, q = 0.041, Figure 4). The abundance of one unidentified genus from the family Enterobacteriaceae instead was significantly increased (HC median 0.00%, CE cobalamin deficient 0.42%, q = 0.041, Figure 4). A trend (q < 0.100) towards decreased abundance was also observed for genera Fusobacterium (HC median 32.61%, CE cobalamin deficient 0.54%, q = 0.056, Figure 4) and Roseburia (HC median 0.03%, CE cobalamin deficient 0.00%, q = 0.092).

Dogs with cobalamin deficiency were followed up after 3 months and subdivided by supplementation route (parenteral, PE, or oral, PO). Bacterial richness (Chao1) and evenness (Shannon Index) were reduced in dogs receiving PE supplementation at baseline (*p* = 0.036 and 0.032, respectively) but not in dogs receiving PO supplementation compared to healthy controls (Appendix A). Follow-up samples after 3 months of supplementation were not significantly different from healthy controls. No difference in richness as measured by Observed ASVs was observed (Appendix A).

Overall microbiome composition (beta diversity, measured with Weighted UniFrac distances) at baseline was significantly different for both supplementation groups compared to healthy controls, although the difference was more pronounced with the PE supplementation group (PE R = 0.653 *p* = 0.001, PO R = 0.267 *p* = 0.005), as shown in Figure 5A. The composition of the microbiome in follow-up samples remained significantly different in both dogs receiving PE supplementation (R = 0.420, *p* = 0.013) and PO supplementation (R = 0.251, *p* = 0.007), as shown in Figure 5B.

When individual bacterial taxa were analyzed (Appendix A), major changes were observed at the phyla level compared to healthy controls, with dogs randomized to receive PE supplementation showing significant differences in 4/5 phyla, and dogs in the PO supplementation group showing significant differences in 1/5 phyla, despite similar median values for all phyla as shown in Figure 6. At follow-up, all five phyla were no longer statistically different from healthy controls, despite showing only minimal improvement in median values (Figure 6). Of note, phyla Bacteroidetes and Fusobacteria, which in healthy controls corresponded to 27.92 and 32.61% of sequences, remained extremely low in both dogs receiving PE supplementation (0.65 and 0.37%, respectively) and in those receiving PO supplementation (2.79 and 3.49%, respectively).

At the genus level, despite randomization at enrollment, only dogs in the PE group had a significantly decreased abundance of Bacteroides (HC median 22.61%, PE baseline 0.03%, q = 0.004), and Turicibacter (HC median 0.28%, PE baseline 0.00%, q = 0.016, Figure 7). The abundance of Helicobacter (HC median 0.64%, PE baseline 0.00% q = 0.007, PO baseline 0.01% q = 0.031) was decreased in both PE and PO at baseline. The abundance of Sutterella (HC median 1.43%, PE baseline 0.01% q = 0.097, PO baseline 0.06% q = 0.099) showed a trend toward a decrease in both PE and PO at baseline, which was also observed for Fusobacterium (HC median 36.61%, PE baseline 0.33%, q = 0.058, Figure 7). In contrast, one unidentified genus from the family Enterobacteriaceae instead was significantly increased at baseline only in the group randomized to receive PO supplementation (HC median 0.00%, PO baseline 1.63%, q = 0.040, Figure 7). No significant differences from healthy controls were observed in the 3-month follow-up samples for either supplementation route.

## 4. Discussion

This study aimed to answer two main questions: one, is cobalamin deficiency in dogs with CE associated with a different dysbiosis profile? Two, does cobalamin supplementation, whether oral or parenteral, have an impact on microbiome composition? For that purpose, we enrolled 47 dogs diagnosed with CE, 18 of which had normal serum cobalamin levels and 29 that had cobalamin deficiency. Ten healthy dogs were recruited as controls. Dogs with cobalamin deficiency were randomized to receive cobalamin by either oral (PO, 18 dogs) or parenteral (PE, 11 dogs) route and were reanalyzed after 3 months of supplementation. Three months of follow-up samples were available for 15/18 dogs in the PO group and 5/11 dogs in the PE group. Clinical scores did not differ between dogs with cobalamin deficiency or normocobalaminemia at baseline.

Cobalamin deficiency was associated with changes in microbiota composition which included decreased richness and changes in beta diversity compared to both healthy controls and CE dogs with normocobalaminemia. Interestingly, no differences in richness or beta diversity were observed between CE dogs with normocobalaminemia and healthy controls. Because previous studies in CE did not separate patients by cobalamin levels, it is difficult to compare our results with those studies. However, our findings in the CE group with cobalamin deficiency match previous findings in canine CE, including decreased richness and evenness [45,46] and changes in beta diversity [23,45,46].

Similarly, findings at the phylum level for the CE group with cobalamin deficiency match previous studies with dogs with CE, with increased Firmicutes and Actinobacteria and decreased Bacteroidetes and Fusobacteria [23]. Similar trends were observed in the CE normocobalaminemic group, but to a lesser extent, and did not reach statistical significance. Those findings suggest that a similar underlying pathological process may be present in CE regardless of cobalamin serum levels but are aggravated by cobalamin deficiency. Since no difference in CIBDAI scores was observed, it is unlikely that those differences in microbiome composition reflect an increase in clinical severity.

In our study, increases in Firmicutes were driven mostly by unidentified species within the family Clostridiaceae (HC median 6.13%, CE non-def median 9.38%, CE CBL def median 22.27%), which was, however, no longer significant once the *p*-value was adjusted for multiple comparisons (q = 0.133). Clostridiaceae is a large family that includes both pathogens, such as *Clostridium perfringens*, and beneficial bacteria, such as *Clostridium hiranonis*, making it difficult to establish the significance of this finding. However, increased Firmicutes to Bacteroidetes ratio observed in dogs with CE and cobalamin deficiency is a common finding in dysbiosis across species and has been described in GI- [47,48] and non-GI-related [49,50] diseases, including canine CE [23].

The production of short-chain fatty acid (SCFA) is known to be depleted in dogs with CE [51]. Bacteroidetes, a phylum found to be significantly decreased in dogs with CE with cobalamin deficiency at baseline, contains genera, such as *Bacteroides* spp. and *Prevotella* spp., that are SCFA-producers. Genus Bacteroides was significantly decreased at baseline in dogs with CE and cobalamin deficiency. When randomized for inclusion in one of the supplementation groups, phylum Bacteroidetes was significantly decreased in both groups compared to healthy controls at baseline, but the decrease in genus Bacteroides only reached significance in the PE group (PE q = 0.004, PO q = 0.086). Interestingly, despite the lack of improvement in median values at the 3-month follow-up, neither at the phylum nor the genus level significance could be found. While oral supplementation with cobalamin has been described to decrease Bacteroides abundance in mice [20], we did not observe that effect in our cohort. Unfortunately, SCFAs were not measured in this study, but it is likely that SCFA production was impaired in agreement with a previous study in canine CE [51] and was not completely restored by cobalamin supplementation.

Fusobacteria is another major phylum in the fecal microbiome of healthy dogs [24], whose abundance was severely decreased in CE dogs with cobalamin deficiency at baseline compared to healthy controls (CE CBL def median 0.54%, HC median 32.61%, q = 0.014), which is in agreement with the literature for CE [25,30,51]. Some Fusobacterium species are known to produce butyrate, an SCFA, from amino acids, which could explain its role in the healthy microbiome of dogs [24,52]. While Fusobacteria was no longer significantly different after 3 months of cobalamin supplementation in both groups, its median abundance remained 10 and 100-fold below that of healthy controls with PO and with PE supplementation, respectively (PO 3 months median 3.49%, q = 0.280; PE 3 months median 0.37%, q = 0.275).

Proteobacteria, and in particular γ-Proteobacteria, are typically increased in dogs with CE [24]. Gamma-Proteobacteria are mainly composed of Enterobacteriaceae (e.g., *E. coli*), and their increase is a hallmark of dysbiosis which has been associated with a number of diseases. In contrast, in our study, we found a significant decrease in Proteobacteria in CE dogs with cobalamin deficiency at baseline compared to healthy controls, driven by a depletion of β-Proteobacteria from genus Sutterella, in agreement with a previous study on immune-suppressant-responsive enteropathy [23]. A significant increase in family Enterobacteriaceae was observed in CE dogs with cobalamin deficiency at baseline compared to healthy controls, driven by an unidentified genus. Both the decrease in Sutterella and the increase in Enterobacteriaceae were no longer significant at the 3 months follow-up.

Our study has limitations that need to be considered. Due to the clinical nature of the study, we enrolled dogs prospectively based on their serum cobalamin levels and did not exclude animals based on their response to treatment. Therefore, our cohort includes animals that were ultimately classified as food-responsive, immune-suppressant-responsive, antibiotic-responsive, and non-responsive, including one case of protein-losing enteropathy. While no study has been able to identify differences in microbiome composition in those subtypes, it is possible that minor differences could be confounding factors in the study.

Another limitation was that, despite our best efforts of randomization, there were small differences at baseline between the two groups receiving cobalamin supplementation (PO vs. PE). This limitation, combined with the lower number of animals at the 3-months follow-up, has limited our statistical power to compare the response of the microbiome to different routes of supplementation. Despite that, we could demonstrate that cobalamin supplementation for 3 months resulted in minimal changes in microbiome composition in terms of alpha diversity, beta diversity, and taxonomy, regardless of administration route. It is possible that supplementation over a longer period of time may be required to modify microbiome composition, in line with findings of a previous study in which dogs with CE treated with steroids only improved microbiome composition in a long-term (1-year) follow-up [23].

Lastly, serum MMA concentrations were not available for CE dogs with normocobalaminemia. Even though there was no significant difference in serum cobalamin concentrations between the healthy group and CE dogs with normocobalaminemia, approximately 10% of dogs with serum cobalamin concentrations in the same range had supranormal serum MMA concentrations in one study [53]. Consequently, there is a risk that a minority of the CE dogs with normal serum cobalamin concentrations had intracellular cobalamin deficiency.

## 5. Conclusions

In conclusion, despite a significant difference in microbiome composition between CE dogs with cobalamin deficiency or normocobalaminemia, cobalamin is unlikely to be the culprit of those differences. Cobalamin supplementation, in combination with appropriate therapy, failed to restore the microbiome composition in our study regardless of the administration route. Serum cobalamin is likely an indicator of differences in underlying pathophysiology that do not influence clinical severity but result in a significant aggravation of dysbiosis. These findings prompt the need for further investigation into the role of the microbiome in cobalamin deficiency associated with CE in dogs.

## Figures and Tables

**Figure 1 animals-13-01378-f001:**
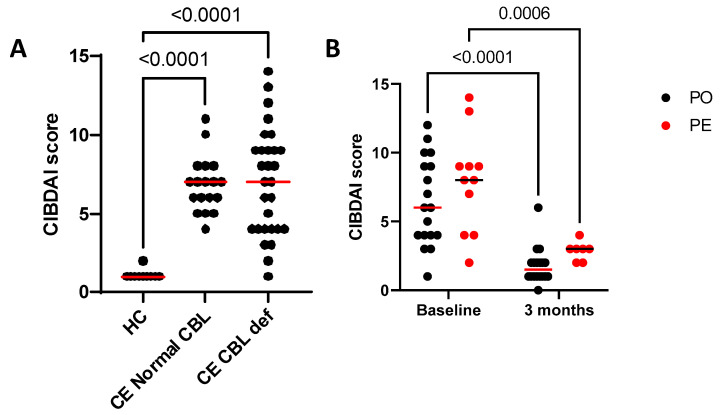
Canine Inflammatory Bowel Disease Activity Index (CIBDAI) scores from (**A**) chronic enteropathy (CE) dogs with cobalamin (CBL) deficiency (CE CBL def) or normocobalaminemia (CE Normal CBL) at inclusion, compared to healthy controls (HC). Scores were significantly higher in dogs with CE CBL def and CE Normal CBL compared to HC (*p* < 0.0001 for both). No difference was observed between CE CBL def and CE Normal CBL (*p* = 0.99). (**B**) CIBDAI scores from CE dogs with cobalamin deficiency at inclusion and after 3 months of cobalamin supplementation (combined with appropriate therapy on a case-by-case basis), separated by route of administration (oral, PO, or parenteral, PE). No difference was observed between groups at baseline (*p* = 0.273) or at 3 months (*p* = 0.580). CIBDAI scores decreased significantly in both groups at 3 months compared with baseline (PO *p* < 0.0001; PE *p* = 0.0006).

**Figure 2 animals-13-01378-f002:**
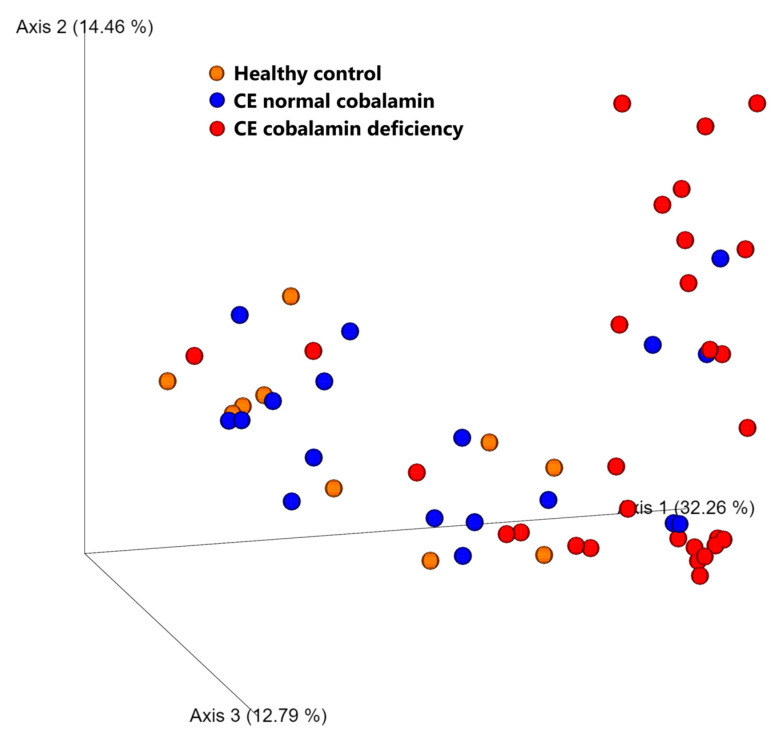
PCA plot of weighted UniFrac distances (beta diversity) of baseline fecal samples from dogs with CE, color-coded based on cobalamin levels (red dots–cobalamin deficient, blue dots–normocobalaminemia), compared to healthy controls (orange dots). Samples from cobalamin-deficient dogs clustered separately from those from normocobalaminemic and healthy dogs. No significant difference was observed between samples from CE dogs with normocobalaminemia and healthy controls.

**Figure 3 animals-13-01378-f003:**
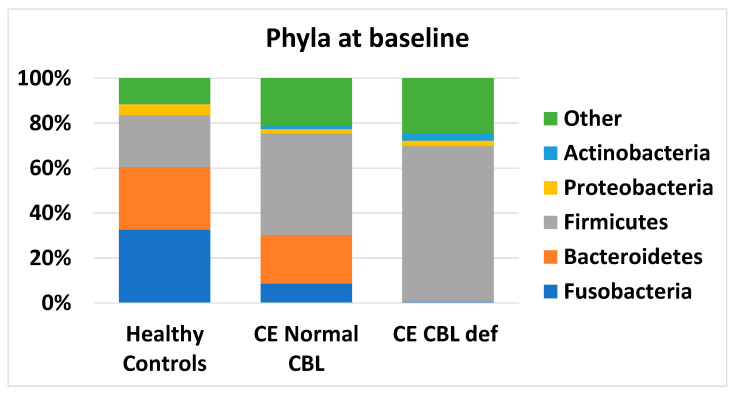
Median abundance of main phyla at baseline in healthy controls, in CE dogs with cobalamin deficiency, and in CE dogs with normocobalaminemia. Values are expressed in percentages of total sequences.

**Figure 4 animals-13-01378-f004:**
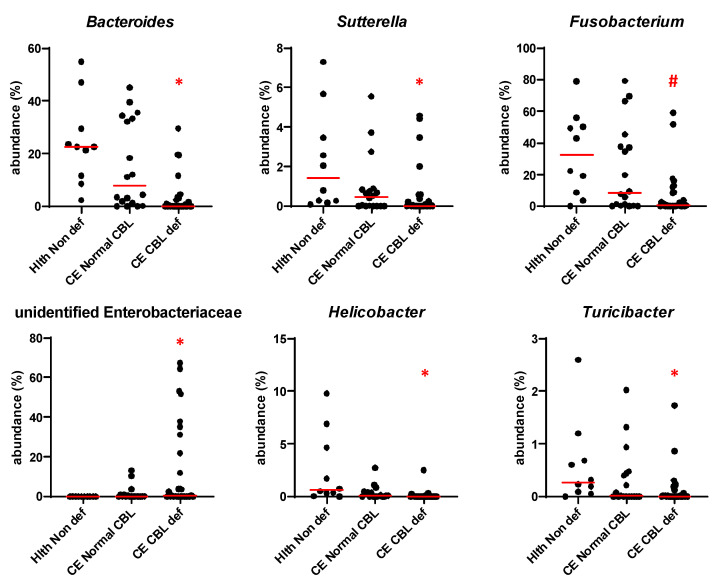
Abundance of key genera at baseline in healthy controls, in CE dogs with cobalamin deficiency, and in CE dogs with normocobalaminemia. Individual values are shown, and red lines indicate the median. * = q < 0.05; # = q < 0.1.

**Figure 5 animals-13-01378-f005:**
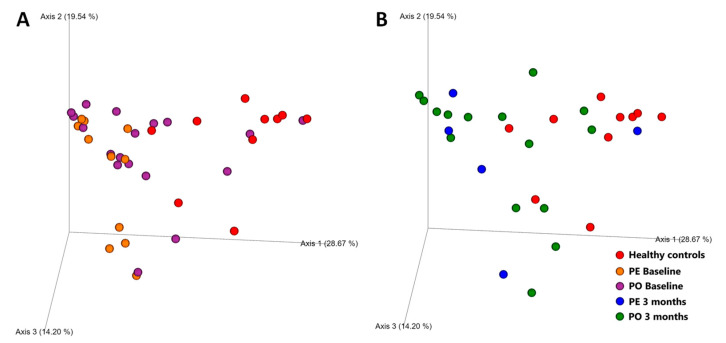
PCA plot of weighted UniFrac distances (beta diversity) of (**A**) baseline fecal samples from dogs receiving cobalamin supplementation parenterally (orange dots) or orally (purple dots), compared to healthy controls (red dots); and (**B**) 3-month follow-up fecal samples from dogs receiving cobalamin supplementation parenterally (blue dots) or orally (green dots), compared to healthy controls (red dots).

**Figure 6 animals-13-01378-f006:**
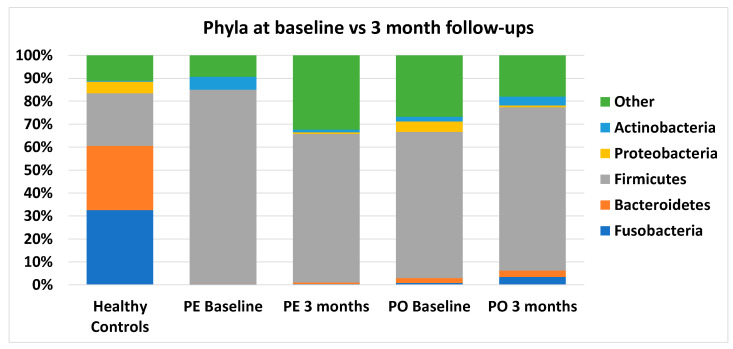
Median abundance of main phyla at baseline in CE dogs with cobalamin deficiency compared to their 3-month follow-ups, separated by route of supplementation (parenteral, PE, or oral, PO) and to healthy controls. Values are expressed in percentages of total sequences.

**Figure 7 animals-13-01378-f007:**
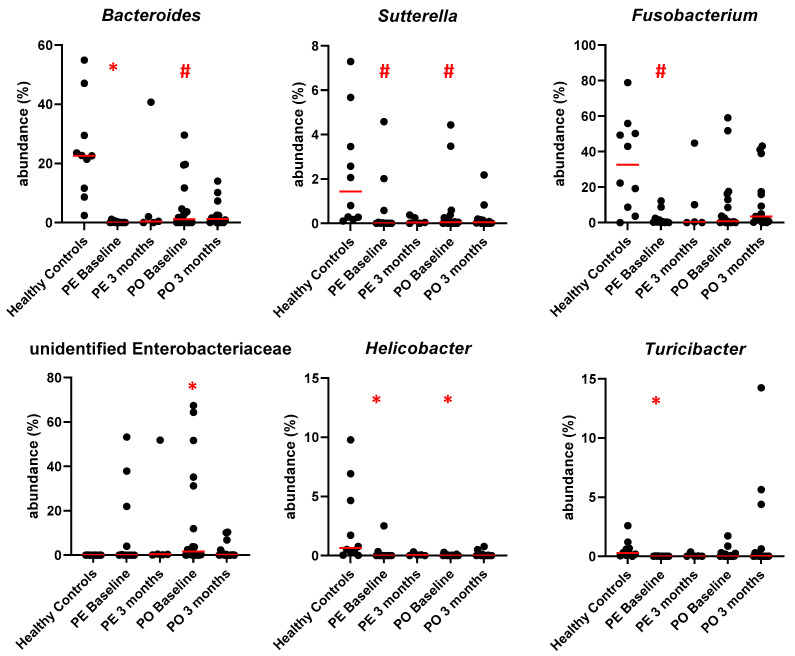
Abundance of key genera in healthy controls and in CE dogs with cobalamin deficiency separated by supplementation route, at baseline and after 3 months of cobalamin supplementation. PE indicates parenteral supplementation; PO indicates oral supplementation. Individual values are shown, and red lines indicate the median. * = q < 0.05; # = q < 0.1.

**Table 1 animals-13-01378-t001:** Selected parameters data from 29 dogs with chronic enteropathy (CE) and cobalamin deficiency, 18 normocobalaminemic dogs with CE, and 10 healthy dogs. Numbers in parentheses indicate the median.

Parameter (Range; Median)	CE + cbl ^a^ Deficiency	CE; Normal cbl	Healthy
Number of dogs	29	18	10
Age (years)	1.5–13.1 (6.1)	0.5–12.5 (5.2)	2.1–10.3 (5.0)
BW ^b^ (kg) at inclusion	4.1–49.0 (11.8)	8.9–32.7 (20.9)	3.6–30.8 (7.6)
BCS ^c^ at inclusion (X/9)	3–7 (4)	2–7 (4)	4–6 (5)
Sex (m/MN/F/FN)	12/6/6/5	11/2/2/3	0/4/4/2
Serum alb ^d^ concentration (g/L); RI ^e^ 29–39	14–37 (30)	27–38 (31)	31–39 (33)
Serum cbl concentration (pmol/L) at inclusion; RI 180–708	<111–210 (183)	350–>738 (492)	350–658 (475)
Serum cbl concentration (pmol/L) after cbl supplementation; RI 180–708	434–2894 (727)	n/a	n/a
Serum MMA ^f^ concentration (nmol/L) at inclusion; RI 414–1193	566–2468 (934)	n/a	470–972 (746)
Serum MMA concentration (nmol/L) at follow-up; RI 414–1193	450–1221 (626)	n/a	n/a
**Diet at inclusion**			
KD ^g^; maintenance diet	8	1	8
KD: ‘Intestinal’	5	8	1
KD; single protein	8	3	1
KD; hydrolyzed	1	6	n/a
KD + Home-cooked	2	n/a	n/a
Home-cooked	5 ^h^	n/a	n/a
**Treatment at inclusion:**			
Corticosteroids	8 ^i^	10 ^j^	n/a
Cyclosporine	3 ^k^	n/a	n/a
Prebiotics	3 ^l^	3 ^m^	n/a
Probiotics	3 ^n^	5 ^o^	n/a
Miscellaneous	9 ^p^	4 ^q^	n/a

^a^ Cobalamin ^b^ Body weight ^c^ Body condition score ^d^ Albumin ^e^ Reference interval ^f^ Methylmalonic acid ^g^ Kibble diet ^h^ One dog was fed a balanced home-cooked diet ^i^ Budesonide 1/8, Methylprednisolon 5/8, Prednisolon 2/8 ^j^ Budesonide 3/10, Methylprednisolon 5/10, Prednisolon 2/10 ^k^ Combined with corticosteroids in 3/3 dogs ^l^ Psyllium husk ^m^ Multi-fiber pellets 2/3, Psyllium husk 1/3 ^n^ SLAB51 2/3, *Enterococcus faecium* (NCIMB10415) 1/3 ^o^ SLAB51 4/5, *Enterococcus faecium* (NCIMB10415) 1/5 ^p^ Olsalazine 4/9, Chaolin clay 2/9, Metoclopramide + Olsalazine 1/9, Maropitant 1/9, Sucralfate 1/9 ^q^ Olsalazine 2/4, Folate 2/4.

## Data Availability

Microbiome raw sequences are available at the NCBI Sequence Read Archive under accession number PRJNA863651.

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
