# Peer review of "The Intestinal Microbiome in Dogs with Chronic Enteropathies and Cobalamin Deficiency or Normocobalaminemia—A Comparative Study"

_animals, 2023, doi:10.3390/ani13081378_

Round 1

Reviewer 1 Report

It is a very interesting manuscript that provides information on the fecal bacterial microbiota in dogs with chronic enteropathies (CE) with cobalamin deficiency, dogs with CE and normocobalaminemic, and a healthy control group of dogs. In addition, dogs with cobalamin deficiency were also assessed after oral or parenteral cobalamin supplementation. The authors found that dogs with a decrease in serum cobalamin presented severe alterations in the bacterial composition of the gut microbiota while those with a normal level did not. In addition, supplementation did not correct these shifts, suggesting that low serum cobalamin levels are an indicator of changes in the bacterial microbiota rather than their cause. This result is very interesting given that cobalamin low serum levels is a common finding in dogs with CE.

The work is well designed, the material and methods section is correct and the results are well presented and well discussed. In my opinion, only some minor corrections should be made.

Abstract

Lines 32-33. The authors could specify the group in which these findings were observed (cobalamin deficiency dogs when compared with the healthy control group).

Introduction

Lines 79 (and 350, 398 and 461): Bacteroides should be in italics (genus).

Line 98: Replace “16srRNA” with “16S rRNA”.

Material and Methods

Line 118-120: According to the reference interval presented (180-708 pmol/l), the dogs with low normal serum cobalamin (also described as the lowest end of the serum cobalamin reference interval [180-210 pmol/l]) are included along with the hypocobalaminemic (<180 pmol/l) inside the group of low serum cobalamin. Thus, it should be as follows: “The group of dogs with CE and low normal serum cobalamin concentrations included dogs with low normal serum cobalamin concentration (180-210 pmol/l; reference interval: 180-708 pmol/l) and dogs with hypocobalaminemia […]”.

Line 121: The period is missing in the sentence before “. All of these dogs […]"

Line 126: There is no space between the sentence and the bibliographic citations “[…] concentrations [32-34].

Line 165: from the from

Line 213: There is no space between “p < 0.05”. Check this in all the manuscript.

Results

Line 247: There is a discordant number in the serum cobalamin concentration at inclusion of the healthy control group when comparing the manuscript and table 1. In the line 247 appears as 350-658 pmol/l (median 475). However, in Table 1, it appears as 350-648 (475) mmol/L. Check the units.

Table 1:

-        Unify the type of hyphen and check the spaces, e.g.: 4– 6 (5); 350- >738 (492).

-        In the text the units of serum cobalamin concentrations were “pmol/l”. However, in table 1, the units are presented as “mmol/L”. Please unify the correct units in the full manuscript and tables.

Line 296: There is no space between the sentence and the bibliographic citation “insignificant_[36].

Line 297: “(p<0.001 for both groups, figure 1)”. However, in figure 1, the p-values shown are p<0.0001.

Lines 339-344: Authors should consider include mentioning the supplementary table 1. These results and their q-values appear in the abstract as an important result and the q-values only appears in this supplementary table.

Lines 349-358: Genera mentioned in this paragraph should be in italics (i.e., Bacteroides, Sutterella, Turicibacter, Fusobacterium and Roseburia).

Line 387: figure 6 instead of 5?

Line 397-404: Genera mentioned in this paragraph should be in italics (i.e., Bacteroides, Turicibacter, Sutterella, Fusobacterium and Roseburia).

Discussion

Line 455: There is an extra space after the word deficiency.

Line 507: with normocobalaminemia_,

Author Response

We appreciate the comments and thank you for your picking up those mistakes. We have made all suggested minor corrections in the text.

Reviewer 2 Report

The number of dogs involved in the study is very small, particularly in the cobalamin supplementation part of the study. Results are interesting, but not definitive. 3-month follow-up samples were available for only 5/11 dogs receiving parenteral cobalamin. Why so few? 

The duration of cobalamin therapy was quite short, only 3 months. A longer treatment period is needed to determine clinical and microbiome effects.

Antibiotics were not allowed, but immunosuppressive treatments. Why were raw-fed dogs not admitted to the study?

Author Response

We thank the reviewer for the questions. Please find our detailed responses below:

Question: The number of dogs involved in the study is very small, particularly in the cobalamin supplementation part of the study. Results are interesting, but not definitive. 3-month follow-up samples were available for only 5/11 dogs receiving parenteral cobalamin. Why so few? 

Answer: Unfortunately we had to exclude several dogs due to confounding factors that could affect microbiome composition. Because this was a clinical study with client-owned dogs, it would be unethical to withhold medication needed to treat concurrent conditions, so dogs were allowed to receive whatever medication was deemed necessary by the attending clinician, however that meant we had to exclude animals from the study, or if medication was started after the collection of the baseline sample, the animal was included in the baseline analysis but not the follow-up. For the follow up samples, as mentioned in lines 317-320: “Reasons for lack of a follow-up sample were drop-out from the study for unknown reasons (2/9), euthanasia due to poor response to medical treatment (2/9), the dog owner for-getting to bring a fecal sample (2/9) or the dogs started treatment with antibiotics (2/9) or proton-pump inhibitor (1/9) prior to follow-up.” While we do not have enough animals to make any strong conclusions about the effect of the different routes of supplementation, the observed effect over the 20 dogs that received supplementation for 3 months, regardless of the route, and were included in the study clearly indicated a lack of changes in the microbiome, which suggests that the underlying cause was not merely the lack of cobalamin.

Question: The duration of cobalamin therapy was quite short, only 3 months. A longer treatment period is needed to determine clinical and microbiome effects.

Answer: We appreciate the suggestion to investigate longer follow-up times, and it is certainly something we will look into for future studies. However, even the short-term effect of supplementation hadn’t been described in dogs before, and therefore we consider these results as a first step towards addressing this lack of knowledge.

Question: Antibiotics were not allowed, but immunosuppressive treatments. Why were raw-fed dogs not admitted to the study?

Answer: Raw-fed dogs were not allowed in the study due to the well-described impact on microbiome composition, which would be a confounding factor in this study.

Reviewer 3 Report

The article is a very comprehensive work, on which I have only minor comments. The key words should not repeat the words used in the title of the article. In the results section, lines 245-248 and 255-258 repeat values that are in the table, so it is redundant. The use of numerical indices in Table 1 is somewhat confusing with references , would it not be more appropriate to use letter indices?
There is no need to repeat the abbreviation CIBDAI on line 291, it is already used for the first time on line 167.

Author Response

We thank the reviewer for the comments. We have changed the keywords and the numerical indices, and removed the repeated abbreviation. While lines 245-248 and 255-258 do repeat values from the table, they are necessary for the flow of the text as we build up information on them. We would prefer to keep the values readily available on the table as well, but are open to removing them from the table if necessary.